# Validation of a Digital Headache Calendar at a Tertiary Referral Center

**DOI:** 10.3390/diagnostics14010021

**Published:** 2023-12-22

**Authors:** Amalie Kjerrumgaard, Jeppe Hvedstrup, Louise Ninett Carlsen, Kristine Dyre, Henrik Schytz

**Affiliations:** 1Danish Headache Center, Department of Neurology, Copenhagen University Hospital—Rigshospitalet-Glostrup, 2600 Glostrup, Denmark; jeppe.hvedstrup.mann@regionh.dk (J.H.); louise.ninett.carlsen@regionh.dk (L.N.C.); henrik.winther.schytz.01@regionh.dk (H.S.); 2Center for IT and Medical Technology, Department Patient at Home, Team Prevention and Outpatient Treatment, The Capital Region of Denmark, 2100 Copenhagen, Denmark; kristine.dyre.02@regionh.dk

**Keywords:** migraine, tension-type headache, digital calendar, validation study

## Abstract

Background: Headache calendars are essential tools in monitoring changes in headache frequency and type. They are used in clinical practice for long-term monitoring, but their validation remains limited. The aim of this study was to validate the use of a digital calendar in monitoring single migraine and tension-type headache attacks. Methods: From July 2022 to February 2023, patients diagnosed with migraine and tension-type headache were enrolled. The validation of the digital calendar involved the comparison of self-reported single headache attacks in the digital calendar with the diagnostic headache diary based on the ICHD-3 criteria for migraine and tension-type headache. Sensitivity and specificity were calculated to assess reliability, and the level of agreement was calculated using kappa statistics. Results: This study included 30 patients (87% women) diagnosed with migraine and tension-type headache. The level of agreement in the classification of a single migraine attack revealed a sensitivity of 82% and a specificity of 78%, representing a substantial level of agreement (κ = 0.60). The classification of a single tension-type headache attack revealed a sensitivity of 84% and a specificity of 72%, with a moderate level of agreement (κ = 0.54). Conclusions: The digital calendar proves effective in monitoring single headache attacks in patients with migraine and tension-type headache. In clinical practice, we recommend using the digital calendar as a monitoring tool for headache patients, as they can accurately identify true migraine and tension-type headache days.

## 1. Introduction

Headache disorders are among the most prevalent and disabling conditions globally, with a prevalence of 52% for headache disorder, 14% for migraine and 26% for tension-type headache (TTH) [1]. Headache calendars are highly valuable and necessary tools in monitoring headache changes in frequency and type and are recommended for headache management [2,3]. Diagnostic headache diaries are useful in the diagnosing of primary headache disorders [4] and have been previously validated in various studies for diagnosing migraine and tension-type headache (TTH) [5,6,7,8]. In the long-term management of preventive treatment in patients diagnosed with a primary headache disorder, a headache calendar is recommended, if patients have learned to distinguish between different headaches [2]. These headache calendars, in which patients only state whether they have experienced migraine or TTH as well as the intensity, are very frequently used in clinical practice and studies [9,10,11] but have not been properly validated. Hence, it is not yet known if days reported as a migraine or TTH day in a headache calendar fulfill the criteria for migraine or TTH [12].

With a shift towards digital health technologies, digital headache calendars have become increasingly popular in the management of headaches because they offer several advantages over traditional paper diaries. Paper diaries cannot be read remotely by a physician and can, from our clinical experience, often be lost or forgotten. One study found that patients with medication-overuse headaches (MOHs) preferred a digital calendar when monitoring headaches and found it to be more convenient than paper calendars [13]. Another study found that among regular users of a digital calendar, the frequency of headache and migraine actually improved over time [14], while it was also shown that patients with chronic pain were found to be more compliant with a digital calendar compared to a paper calendar [15]. Digital calendars allow patients to record and track their symptoms and medication in real time and support physicians in the treatment and management of headaches. However, the majority of available digital calendars lack validation [16]. The aim of this study was to validate the use of a digital headache calendar in monitoring single headache attacks in patients with migraine and TTH at our tertiary referral center.

## 2. Methods

### 2.1. Study Population and Study Design

Patients with migraine and TTH followed at the Danish Headache Center were recruited from July 2022 to February 2023. All patients had previously been diagnosed by a headache specialist at the Danish Headache Center. All included patients were requested to use a digital headache calendar and a paper headache diary simultaneously for a period of 28 days. One being the diagnostic paper headache diary (Appendix A) [8], and the other being a digital calendar, which could be assessed through a digital app named My Healtcareplatform (in Danish: Min Sundhedsplatform (MinSP)). Patients who did not complete both diaries were excluded from the study. This study was approved by the Neuroscience Center at Rigshospitalet-Glostrup as a quality assurance study and approved as such by the Regional Ethics Committee (H-18008942), which determined that participant consent was not required as the study collected information from patients undergoing usual treatment and evaluation. This study was conducted in accordance with Danish Data Laws.

### 2.2. The Diagnostic Headache Diary

The diagnostic headache diary was used to classify single headache attacks according to the International Classification of Headache Disorders third edition (ICHD-3) as either migraine or TTH [12]. When defining a migraine day, we applied the guidelines of the International Headache Society (IHS), stating the definition of a migraine day in clinical trials [10]. Consequently, a migraine day was defined as a day with a headache lasting at least 30 min without intake of analgesics, and meeting ICHD-3 criteria for migraine or probable migraine [10]. 

According to ICHD-3, a definite diagnosis trumps a probable diagnosis, and the general rule of hierarchy puts migraine before TTH [12]. The exception was patients diagnosed only with TTH; here, a single headache attack could not be characterized as a migraine day.

### 2.3. The Digital Headache Calendar (Digital Calendar)

The digital headache calendar (MinSP Hovedpinekalender) was launched on 24 May 2022, and can be assessed using MyChart by Epic via smartphones and web browsers. The digital calendar is then linked to an electronic healthcare record (EHR) system by Epic, which can be accessed by hospital staff. The digital calendar tracks the time and date of entries and the number of entries and registered headache days per entry. 

When developing the digital headache calendar, it was considered that the design should be user-friendly for patients with migraine and TTH. Consequently, the patients were asked only to answer four simple questions: (1) Specify the average intensity of your tension-type headache in the last 24 h (mild/moderate/severe), (2) Specify the average intensity of your migraine headache in the last 24 h (mild/moderate/severe), (3) Have you taken any kind of analgesics for your headache or any other reason in the last 24 h (yes/no), (4) Choose the type of analgesics used in the last 24 h (paracetamol/NSAID/Triptans/Morphine). To assist patients in differentiating between headaches, the digital calendar contained an information box with a short overview of the diagnostic criteria for migraine and TTH (Appendix A). Patients received a notification twice a week (Wednesday and Sunday) to complete the digital calendar. The data entered in the digital calendar were visible to the physician at the clinic, and an overview of headache days and medication use was presented. 

### 2.4. Validation of Digital Calendar

The diagnostic headache diary and the digital calendar were compared to examine whether a single headache attack, registered as migraine or TTH in the digital calendar, would meet the corresponding ICHD-3 criteria in the diagnostic headache diary. We validated the digital calendar using two sets of relevant definitions: (1) migraine and TTH days were classified using the criteria for definite migraine and TTH; (2) migraine and TTH days were classified using the criteria for definite migraine and TTH, as well as being classified as probable migraine and probable TTH.

The patients were asked only to record days with headaches, thus assuming they were headache-free on days no data were entered. If a headache attack was recorded in only one of the diaries, that specific attack would be excluded from the data. If two types of headaches (migraine and TTH) were reported on the same day in the digital calendar, they would both be compared to the characteristics of the headache reported in the diagnostic headache diary and classified accordingly as either migraine or TTH.

### 2.5. Data Analysis

Data are presented as numbers (percentage), mean ± standard deviation (SD) or median and interquartile range (IQR). Group comparisons were made using the Chi-square test, Fisher’s exact test or *t*-test, as appropriate. To evaluate and compare the patients’ ability in identifying headache days, a cut-off threshold of greater than 75% accuracy was selected to determine those with the highest number of correctly identified days. A *p*-value < 0.05 was considered statistically significant. Sensitivity, specificity, positive predictive value (PPV) and negative predictive value (NPV) were calculated as measures of reliability. We calculated the unweighted kappa coefficient (κ) to compare the patients’ classification of single headache attacks to the classification of the diagnostic headache diary. The patients were all estimated together as one counterpart. The agreement of κ was evaluated using the following ranges: moderate: 0.40 ≤ κ < 0.60, substantial: 0.60 ≤ κ < 0.80, excellent agreement: 0.80 ≤ κ ≤ 1.0. Statistical analyses were performed using Microsoft Excel^®^ version 16.72 (© 2023 Microsoft Corporation, One Microsoft Way, Redmond, WA, USA) and RStudio version 1.3.1093 (© 2020 RStudio, PBC, Boston, MA, USA) for macOS.

## 3. Results

### 3.1. Baseline Characteristics

Out of 59 patients included, 30 completed both diaries, while 29 patients were excluded due to incomplete diaries (Figure 1). The patients had a mean age of 41 ± 12 years and 26 (87%) were women (Table 1). In total, 16 (53%) patients were diagnosed with both migraine and TTH. Further baseline characteristics are presented in Table 1. 

The median of self-reported headache days was 15.5 (9.25–21.75) days/month (28 days). Patients reported 8.5 (4.75–13.0) days with migraine and 10.5 (7.25–15.25) days with TTH in the digital calendar. Analgesics were used on 7.0 ± 3.4 days/month. We found that a median of 2.9 (1.7–5.7) headache days were reported in the digital calendar per entry (Table 1).

### 3.2. Validation of Electronic Headache Diary (Digital Calendar)

The level of agreement in the classification of a single headache attack by the diagnostic headache diary and the patients’ self-reported outcome in the digital calendar is shown in Table 2 and Table 3. The different definitions were analyzed including definite migraine and TTH (Table 2A and Table 3A) and probable migraine and TTH (Table 2B and Table 3B). 

#### 3.2.1. Migraine Days

The diagnostic headache diary classification of a single headache attack as definite migraine and the patients’ self-reported migraine resulted in 173 days correctly classified as migraine in the digital calendar. Disagreement between the self-reported outcome in the digital calendar and the diagnostic headache diary resulted in a total of 113 days, of which 86 days were reported as migraine, but did not meet the criteria, and 27 days were reported as a non-migraine headache but did meet the criteria for migraine (Table 2A). The data revealed a sensitivity of 87% and specificity of 62%. We calculated the unweighted kappa coefficient, κ = 0.49, which represented a moderate level of agreement (Table 2A). When including the criteria for probable migraine, the number of correctly identified migraine days increased to 225 days and resulted in a sensitivity of 82% and a specificity of 78%. The unweighted kappa coefficient, κ = 0.60, represented a substantial level of agreement (Table 2B). 

**Table 2 diagnostics-14-00021-t002:** (A) 2 × 2 tables comparing self-reported migraine days in the digital calendar with definite migraine days recorded in the diagnostic headache diary. (B) 2 × 2 tables comparing self-reported migraine days in the digital calendar with definite and probable migraine days recorded in the diagnostic headache diary.

A					B				
		Migraine Days					Migraine Days		
		Yes	No				Yes	No	
Self-reported migraine days	yes	173	86	259	Self-reported migraine days	yes	225	34	259
	no	27	143	170		no	48	122	170
		200	229	429			273	156	429
Sensitivity: 173/200 = 0.87					Sensitivity: 225/273 = 0.82				
Specificity: 143/229 = 0.62					Specificity: 122/156 = 0.78				
PPV: 173/259 = 0.67					PPV: 225/259 = 0.87				
NPV: 143/170 = 0.84					NPV: 122/170 = 0.72				
Unweighted kappa coefficient (κ): 0.49					Unweighted kappa coefficient (κ): 0.60				

#### 3.2.2. Tension-Type Headache (TTH) Days

The diagnostic headache diary classification of a single headache attack as definite TTH and the patients’ self-reported TTH resulted in 164 days correctly classified as TTH. Disagreement between the self-reported outcome in the digital calendar and the diagnostic headache diary resulted in 27 days reported incorrectly as non-TTH headache, and 86 days were reported as TTH, but did not meet the criteria (Table 3A). The data revealed a sensitivity of 86% and specificity of 70%. The unweighted kappa coefficient, κ = 0.52, represented a moderate level of agreement (Table 3A). When including probable TTH, the number of headache attacks correctly identified increased to 173 days. It resulted in a sensitivity of 84% and specificity of 72%. The unweighted kappa coefficient, κ = 0.54, represented a moderate level of agreement (Table 3B).

**Table 3 diagnostics-14-00021-t003:** (A) 2 × 2 tables comparing self-reported TTH days in the digital calendar with definite TTH days recorded in the diagnostic headache diary. (B) 2 × 2 tables comparing self-reported TTH days in the digital calendar with definite and probable TTH days recorded in the diagnostic headache diary.

A					B				
		TTH Days					TTH Days		
		Yes	No				Yes	No	
Self-reported TTH days	yes	164	86	250	Self-reported TTH days	yes	173	77	250
	no	27	199	226		no	32	194	226
		191	285	476			205	271	476
Sensitivity: 164/191 = 0.86					Sensitivity: 173/205 = 0.84				
Specificity: 199/285 = 0.70					Specificity: 194/271 = 0.72				
PPV: 164/250 = 0.66					PPV: 173/250 = 0.70				
NPV: 199/226 = 0.88					NPV: 194/226 = 0.86				
Unweighted kappa coefficient (κ): 0.52					Unweighted kappa coefficient (κ): 0.54				

### 3.3. Comparison between Patient Groups

From a total of 30 headache patients, 14 patients had ≥75% correctly classified single headache attacks (correctly classified as migraine or TTH) and 16 patients had <75% correctly classified single headache attacks (Table 4). When comparing the characteristics of the two groups, we found no significant difference between the age (39.1 ± 13.5 versus 42.9 ± 10.9; *p* = 0.40) or gender (women *n* = 14 versus *n* = 12; *p* = 0.1). We found no significant difference between the prevalence of different headache diagnoses or the number of years with diagnoses between the two groups.

## 4. Discussion

This study investigated the validity of a digital calendar as a tool for monitoring headaches in patients with migraine and TTH and demonstrated that the digital calendar had a high sensitivity in correctly classifying migraine days, regardless of the definitions used. When applying the second criteria in the definition of a migraine day (including both definite and probable migraine), the specificity was increased too, with a substantial level of agreement. This study also examined the reliability of the digital calendar in classifying TTH days (definite and probable TTH) and found a moderate level of agreement. The findings indicate that the digital calendar was optimal in classifying migraine attacks compared to TTH attacks.

This study found that, when using the digital calendar, patients tend to underestimate the number of migraine days, when probable migraine is included. This could be due to the information box in the digital calendar not containing information on probable diagnosis. This study suggests that there might be an overestimation of TTH days due to some patients reporting both TTH and migraine on the same day in the digital calendar, which usually resulted in classification as migraine when compared to the diagnostic headache diary. 

This study also investigated whether there were significant characteristics associated with patients who had a high correct percentage of headache days (≥75%) compared to those with a lower correct percentage (<75%). However, no significant differences were found between the two groups. Perhaps a larger sample size and inclusion of probable diagnosis could potentially reveal significant differences between the two groups, which could be useful in identifying patients where a digital calendar is not sufficient in classifying headache days.

Most other studies have focused on validating the use of headache diaries for diagnosing primary headaches, rather than monitoring single headache attacks in a calendar long-term. However, a study by Roesch et al. [17] classified 102 single headache attacks from an electronic headache diary by both a neurologist and an automated algorithm based on the ICHD-3 criteria for migraine and TTH.

The purpose was to validate the algorithm for the automated classification of migraine and TTH days in an electronic headache diary, and they found a substantial agreement with a kappa value of 0.74 [17]. In comparison, the present study used a simpler digital calendar, without an automated algorithm, but relying on the patient’s ability to differentiate, and found a moderate to substantial level of agreement. The present study’s digital calendar was considered user-friendly with only four questions, which could enhance compliance and make it a valuable tool in headache management.

As previous studies have shown, the use of a daily headache diary may reduce recall bias and increase the reliability of a patient’s reported headache characteristics [2,18,19,20]. This is particularly important when optimizing headache treatment. Compared to paper diaries, digital calendars offer the advantage of allowing patients to promptly record headaches wherever they are, thereby reducing recall bias. A study by Van Casteren D. et al. [21] assessed the value of digital calendars and recall bias in clinically diagnosed patients with migraine by comparing self-reported estimates of monthly migraine days to numbers based on a daily digital calendar. The patients had to recall migraine days in the past three months. They found that self-estimated migraine days tended to be underestimated when less than eight headache days were reported and overestimated when more than eight days were reported. This issue was limited in the present study by the feature of time stamps in the digital calendar, following recommendations from the clinical guidelines [10]. The findings revealed a median of 2.9 headache days reported per entry in the digital calendar, suggesting that most patients were regularly reporting their headaches, thereby reducing potential recall bias. It is presumed that patients recorded headaches in both the digital calendar and the diagnostic headache diary simultaneously. However, since the diagnostic headache diary was a paper-based format, patients may have completed multiple entries at the same time, making it susceptible to recall bias.

In recent years, CGRP antibodies have proven to be effective in preventing both episodic and chronic migraines [22]. In Denmark, the Danish Medicines Agency has established national criteria for the administration of CGRP antibodies, limiting its recommendations to patients with chronic migraines due to the high costs associated with treatment [23]. Consequently, the significance of using a digital headache calendar, which minimizes recall bias, becomes even more apparent as it helps patients to accurately document their headaches and qualify for CGRP antibody treatment. The digital calendar can be helpful in recording the frequency and severity of headaches, making it essential for monitoring the effectiveness of headache interventions, such as preventive medication.

### Strengths and Limitations of the Study

This study represents clinical practice and validates the use of a simple digital calendar for monitoring migraine and TTH headaches that will be of high importance in the clinical setting and in research. This study was conducted at a tertiary referral center, which added clinical value but may limit generalizability to other healthcare settings. The patients being monitored and treated at a specialized clinic could be more aware of symptoms, and better at differentiating headaches. The limitations of this study include the small sample size and the patients’ use of the digital calendar for only one month. Long-term consistent use of the digital calendar may yield different data, and when used alone without the accompanying diagnostic paper dairy, which is its intended purpose, patients may not be as precise in distinguishing between headache types. To address this, the digital calendar includes an information box. However, it remains uncertain whether patients will consistently use it. We consider this a limitation of this study but also a reflection of a real-life setting. An additional limitation arises from the IHS guidelines, which specify that a headache can be classified as a migraine day if it responds favorably to acute treatment with a medication specifically designed for migraines [10]. However, our digital calendar does not include information on the response to acute medication, limiting our ability to assess this aspect. Due to this limitation, this study might have underestimated the migraine days. Since the IHS has not defined a TTH day, the sole definition utilized was the criteria outlined in the ICHD-3. It is important to note that relying solely on the ICHD-3 criteria may pose a limitation, as these criteria serve as diagnostic guidelines rather than being designed for individuals already diagnosed with TTH. 

The use of the kappa value in this study had limitations, as the patients were treated as a single counterpart, which deviated from the typical use of the kappa value involving comparisons between two persons or methods. Additionally, only two patients with TTH did not have migraine and specifically patients diagnosed with CTTH were much underrepresented, which could limit the generalizability of the findings regarding TTH. 

## 5. Conclusions

The digital calendar was found to be effective in monitoring single headache attacks in patients with migraine and TTH, when the appropriate diagnostic criteria were applied. The sensitivity for migraine and TTH was 82% and 84%, respectively, with a specificity of 78% for migraine and 72% for TTH. The digital calendar showed a substantial level of agreement with the diagnostic headache diary when classifying migraine days. These findings confirm the reliability of the digital calendar in monitoring patients with migraine and TTH at a tertiary referral center. However, no significant patient characteristics were identified in relation to their performance in this study.

The digital calendar is a valuable tool for managing migraine and TTH, providing accurate information for physicians to prepare for consultations and make informed decisions regarding medication use. In clinical practice, we recommend using the digital calendar as a monitoring tool for headache patients, as it can effectively identify true migraine and TTH days and assist physicians in treatment planning and medication management. 

## Figures and Tables

**Figure 1 diagnostics-14-00021-f001:**
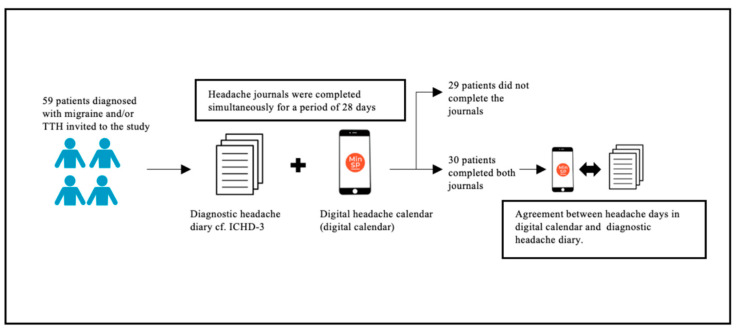
Study flow chart.

**Table 1 diagnostics-14-00021-t001:** Characteristics of patients with headache and self-reported headache days in the digital calendar.

	Headache Patients (*n* = 30)
Age, years mean ± SD	41 ± 12
Women, *n* (%)	26 (87%)
Diagnosis by physician at headache clinic ^1^, *n* (%)	
Episodic migraine	9 (30%)
Episodic TTH	14 (47%)
Chronic migraine	19 (63%)
Chronic TTH	4 (13%)
Years with diagnosis from headache center, *n* (%)	
≤1 year	12 (40%)
2 years	4 (13%)
3 years	5 (17%)
4 years	5 (17%)
5 years	4 (13%)
Headache days in a month (28 days), median (IQR)	15.5 (9.25–21.75)
⇒ Migraine days/month	8.5 (4.75–13.0)
⇒ TTH days/month	10.5 (7.25–15.25)
Days using analgesics ^2^, mean ± SD	7.0 ± 3.4
Number of headache days reported in digital calendar per use, median (IQR)	2.9 (1.7–5.7)

TTH: tension-type headache; data are presented as *n* (%), mean ± standard deviation (SD) or median (IQR). ^1^ A total of 16 patients (53%) were diagnosed with both a migraine and a TTH. ^2^ Including paracetamol, ibuprofen, combination drugs, Triptans.

**Table 4 diagnostics-14-00021-t004:** The characteristics of patient groups were analyzed and compared based on their ability to correctly identify individual headache attacks, using a cut-off threshold of 75% accuracy.

Correctly Identified Headache Attacks, %			
	≥75%	<75%	*p* value
Headache patients (*n* = 30) (%)	14 (47)	16 (53)	
Age, years mean ± SD	39.1 ± 13.5	42.9 ± 10.9	0.40
Women, *n* (%)	14 (100%)	12 (75%)	0.10
Diagnosis by physician at headache clinic, *n* (%)			
Episodic migraine	3 (15%)	6 (23%)	0.71
Episodic TTH	6 (30%)	8 (31%)	0.96
Chronic migraine	9 (45%)	10 (38%)	0.66
Chronic TTH	2 (10%)	2 (8%)	1.00
Years with diagnosis, *n* (%)			
≤1 year	6 (43%)	6 (38%)	0.75
2–4 years	5 (36%)	9 (56%)	0.22
5 years	3 (21%)	1 (6%)	0.96

Data are presented as number (%) or mean ± standard deviation (SD). Comparisons are made using Chi-square test, Fisher’s exact test or *t*-test accordingly. *p*-value < 0.05 was considered statistically significant.

## Data Availability

The data that support the findings of this study are not openly available due to reasons of sensitivity and are available from the corresponding author upon reasonable request.

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
