# Peer review of "Validation of a Digital Headache Calendar at a Tertiary Referral Center"

_diagnostics, 2023, doi:10.3390/diagnostics14010021_

Round 1
Reviewer 1 Report
Comments and Suggestions for Authors
I was not impressed by this research article. The innovation it brings is modest, the sample size is even less so, the selection of the sample itself does not represent the patient population attending a Level 3 Headache Centre (i.e. a disproportionate percentage of CTTH) so I do not consider it useful that an established message (appropriate use of the headache diary) can provide a cultural benefit to readers.
Author Response
Thank you for providing your feedback. We genuinely appreciate the reviewer’s insightful observation regarding the disproportionate representation of chronic tension-type headache (CTTH) patients in our sample, and we will add this to our limitations. It’s important to note that the classification of headache diagnosis, whether chronic or episodic, is depending upon the frequency of headache days and is subject to change over time, particularly in response to headache treatments. Given the real-world clinical context of our study and the voluntary participation of patients, we acknowledge that our sample may not fully encompass the entire spectrum of patients at the headache center. Consequently, generalizing the findings to other healthcare settings may pose challenges, as duly mentioned in the limitations section. We also recognize the limitation posed by the sample size, as indicated in our acknowledgment of study limitations. Your feedback is valuable, and we will duly consider these aspects in refining our study and its implications.
In the Strength and limitations section, page 15 line 7 we added the following sentence:
Additionally, only two patients with TTH did not have migraine and specifically patients diagnosed with CTTH were much underrepresented, which could limit the generalizability of the findings regarding TTH.
Reviewer 2 Report
Comments and Suggestions for Authors
The paper is focussed on the distinction between a "migraine day" and a "TTH day". A paper diary is used as a gold standard so that a digital approach to self-assessment can be compared with the paper diary.
The definitions used in the gold standard are therefore important. "Migraine day" is adequately defined from IHS guidelines paper, but "TTH day" is not defined other than extracting the characteristics from the ICHD3 which is referring to the diagnosis of TTH rather than the features of a single attack.
ICHD3 points out that TTH should not be diagnosed concurrently with chronic migraine: "Because tension-type-like headache is within the diagnostic criteria for 1.3 Chronic migraine, this diagnosis excludes the diagnosis of 2. Tension-type headache or its types." This results in semantic difficulties and requires the use of clumsy terms such as "TTH-like headache" in this context.
Clearly, one benefit of a simple digital self-assessment of headache type would be to avoid the need for a paper diary. In this study, patients had to do both. It is tempting to think that the filling in of the characteristics of the headache on the paper diary may have influenced the patient's decision whether to describe the headache day as migraine or TTH. In other words, perhaps the digital method would be less accurate when used alone. This deserves comment.
The paper states "To assist patients in differentiating between headaches, the digital calendar contained an information box with a short overview of the diagnostic criteria for migraine and TTH." Was this information box displayed before every entry or was reference to it optional? What did the information box say? Its content should be provided (perhaps as supplementary material).
The paper is actually addressing two questions at once, and it is perhaps worth considering them separately:
1. Can patients accurately distinguish between a "migraine day" and a "TTH day" without completing a formal questionnaire of the type used in the written diary?
2. Can a digital diary work as well as (or better than) a paper diary?
The title emphasises the latter, but I think the data is probably more pertinent to the former.
Reviewer 3 Report
Comments and Suggestions for Authors
The authors report a possible validation of a digital headache calendar at a tertiary referral centre. They monitored single migraine and tension-type headache attacks. The data are really interesting; moreover, the manuscript is clear and well written; therefore, I have only few comments for the authors:
- Introduction: In the first para the authors should add data about headache prevalence: the authors must read and cite the paper by Foiadelli T et al. Ital J Pediatr. 2018 Apr Ital J Pediatr. 2018 Apr 4;44(1):44.
- Discussion: the authors should comment the time used to fill headache calendar: this information can be useful for the readers.
Comments on the Quality of English Languagegood
Round 2
Reviewer 1 Report
Comments and Suggestions for Authors
The Authors addressed the raised concerns. The manuscript is now fine with me.